# Delayed Recovery in Idiopathic Sudden Sensorineural Hearing Loss

**DOI:** 10.3390/jcm11102792

**Published:** 2022-05-16

**Authors:** Gina Na, Ki-Won Kim, Keun-Woo Jung, Jimin Yun, Taek-Yoon Cheong, Jeon-Mi Lee

**Affiliations:** Department of Otorhinolaryngology, Ilsan Paik Hospital, Inje University College of Medicine, Goyang 10380, Korea; kalosophiana@gmail.com (G.N.); i9906@paik.ac.kr (K.-W.K.); i9884@paik.ac.kr (K.-W.J.); i9858@paik.ac.kr (J.Y.); i9803@paik.ac.kr (T.-Y.C.)

**Keywords:** sudden sensorineural hearing loss, delayed recovery, steroid therapy

## Abstract

Sudden hearing loss is an easily encountered disease in clinics, but its prognosis has not been completely elucidated. In the present study, we investigated the long-term prognosis of sudden hearing loss with 130 patients who were diagnosed based on strict criteria and provided uniform treatment. The patients with incomplete recovery were reevaluated after 2 months without receiving additional treatment. Hearing levels at different time points were compared. Moreover, the associated factors affecting the degree of hearing improvement over time were evaluated using stepwise multiple linear regression. After treatment, 73 out of the 130 (56.1%) patients attained incomplete recovery and were reevaluated after 2 months. Seventeen out of the seventy-three (23.3%) patients showed a grade promotion, fifty-four (74%) were constant, and two (2.7%) were aggravated. The mean interaural hearing differences (IHDs) showed significant improvement. Old age, poor initial IHD, and poor recovery grade were significantly associated with a profitable delayed hearing gain. Poorer hearing level at the time of onset might be a sign for slower recovery rather than a poorer prognostic factor. The treatment outcome of idiopathic sudden sensorineural hearing loss (ISSNHL) should be evaluated at least 2 months after treatment completion, and counseling is required due to the need for long-term follow-up in patients with ISSNHL.

## 1. Introduction

Idiopathic sudden sensorineural hearing loss (ISSNHL) is observed in 5–20 per 100,000 inhabitants in the United States and is naturally restored in 47–63% of patients within 2 weeks [1]. Although its etiology and pathophysiological mechanisms have not been completely elucidated, several factors, such as age, vertigo, extent of hearing loss, configuration of the audiogram, treatment delay from the onset, neutrophil-to-lymphocyte ratio, and magnetic resonance imaging findings, affect its prognosis [2,3,4,5,6,7,8]. Nevertheless, it is impossible to predict one’s prognosis before treatment. As recommended in the American Academy of Otolaryngology-Head and Neck Surgery (AAO-HNS) guidelines, a clinician managing patients with ISSNHL should be aware of the prognosis and proper time to start rehabilitation. Recent AAO-HNS guidelines have recommended that the recovery course should be observed for up to 6 months [9]. However, patients who have not yet reached a complete recovery demand the information in real clinics and are eager to determine how long to wait.

There have been few studies regarding the long-term prognosis of ISSNHL. Most studies have agreed that some degree of hearing recovery occurs even after the discontinuation of drug administration [8,10]. Although these studies have clearly demonstrated a tendency for delayed hearing recovery, the populations and designs of the studies were significantly diverse to deduce firm and detailed conclusions. For example, Fetterman et al. and Halpin et al. observed a large number of patients with ISSNHL (242 and 250) for 4–6 months and reported a favorable delayed hearing recovery [1,11]. However, the enrollment criteria in both studies were inconsistent with the currently published AAO-HNS criteria, and even patients with objective symptoms were included [1]. Moreover, the treatment protocol was incompatible with the current evidence-based treatment, and the time point for comparing hearing recovery was irregular. Noguchi et al. also observed 142 patients with ISSNHL who were consistent with the AAO-HNS criteria for 13 months [12]. However, the study included patients with vertigo, who might not be strictly patients with ISSNHL. The treatment methods were also not unified; for example, salvage treatments were provided to only a few selected patients without evidence, and treatments other than steroid therapy, such as oxygen therapy, were also provided.

In the present study, we investigated the long-term prognosis of ISSNHL and aimed to determine which factors are associated with delayed recovery. We applied strict diagnostic criteria and provided uniform treatment. This minimized the possibility that the results could be distorted due to nonuniform conditions, as in previous studies. We believe that the present study would help us understand the long-term prognosis of ISSNHL and further assist in patient treatment and counseling.

## 2. Materials and Methods

### 2.1. Participants

We retrospectively reviewed patients who were diagnosed with sudden sensorineural hearing loss (SNHL) and subsequently treated from March 2017 to November 2019. Sudden SNHL was diagnosed by pure-tone audiometry, according to its audiometric criteria, which was a hearing loss of ≥30 dB compared to the opposite ear’s threshold, occurring in at least three consecutive frequencies. The inclusion criteria were as follows: (1) age ≥ 19 years, (2) time from symptom onset to initial corticosteroid therapy ≤ 14 days, and (3) completion of the authors’ protocols for the follow-up schedule, which is described in the “Treatment protocol” session below. The exclusion criteria were as follows: (1) the presence of underlying conditions associated with sudden SNHL, such as the presence of a retrocochlear lesion, infection, trauma, and ototoxic medication; (2) bilateral ISSNHL; (3) recurrent fluctuating hearing loss; (4) SNHL combined with vertigo; and (5) accurate treatment outcomes that cannot be achieved, for example, a lack of audiometric confirmation before treatment or preceding asymmetric hearing loss. Data, including age, sex, the affected side, medical history (hypertension and diabetes mellitus), and time delay to treatment, were collected. This study was approved by the Institutional Review Board of Inje University Ilsan Paik Hospital (approval number: 2021-02-007) and carried out following the principles of the Declaration of Helsinki.

### 2.2. Treatment Protocols

Corticosteroid therapy was offered as an initial treatment. Corticosteroid was administered either orally or by intratympanic injection (ITI) therapy, according to the patient’s medical condition. When administered orally, the patients were treated with oral methylprednisolone (0.8 mg/kg/day) for the first 5 days and then tapered off for 5 days. When treated with ITI therapy, an intratympanic dexamethasone (5 mg/mL) injection was administered six times for 2 weeks. After 10 days of oral corticosteroid therapy or 2 weeks of ITI therapy, a pure-tone audiogram was performed for hearing confirmation. The treatments were interrupted when the complete recovery was confirmed, but salvage therapy was offered when the recovery was incomplete. Salvage therapy included a total of six times for 2 weeks of ITI therapy for all patients. The hearing was confirmed again after 2 weeks of salvage therapy, and no more treatment was further provided for the patients with incomplete recovery. Instead, they were instructed to visit 2 months later to reassess their hearing. During the treatment, no other therapy than corticosteroid was provided, such as hyperbaric oxygen treatment; stellate ganglion block; or other pharmacologic therapies, such as antivirals, thrombolytics, vasodilators, or vasoactive substances.

### 2.3. Audiometric Evaluation and Recovery Assessment

The hearing level was assessed using a pure-tone audiogram before treatment, immediately after treatment, and 2 months after treatment completion. The pure-tone average was defined as the mean value of the measurements taken at the 500, 1000, 2000, and 4000-Hz frequencies. When the hearing thresholds were undetectable, a fixed value of 120 dB was applied.

The hearing recovery grade was classified according to the criteria proposed by the AAO-HNS [9,13] (Table 1). The degree of hearing recovery was analyzed immediately after treatment and 2 months after treatment completion.

We also analyzed the actual hearing gain immediately after treatment and 2 months after treatment completion. We used the interaural hearing difference (IHD) between affected and unaffected ears to estimate the actual hearing loss of the individuals. The change of IHD over time was considered a hearing gain. For example, the hearing gain immediately after the treatment was calculated by subtracting the IHD immediately after treatment from the IHD before treatment. The delayed hearing gain was calculated by subtracting IHD 2 months after treatment completion from the IHD immediately after treatment.

### 2.4. Statistical Analyses

All statistical analyses were performed using GraphPad Prism 8 for Windows (GraphPad Software, La Jolla, CA, USA) and SPSS software version 21 for Windows (IBM Corp., Armonk, NY, USA). Mean and standard deviation were used for descriptive statistics. Grade differences were tested using the chi-squared test.

The IHD over time was analyzed using a repeated measures analysis of variance (RM-ANOVA) with three levels by time (before treatment, immediately after treatment, and 2 months after treatment completion). Wilks’ lambda multivariate test was run to explain the effect of time. The Greenhouse–Geisser test was established when Mauchly’s sphericity did not prove the homogenous data. Moreover, we performed Bonferroni post-correction to compare differences over time.

In the meantime, stepwise multiple linear regression predicted the associated factors for the change of IHD over time, delayed hearing gain, as a dependent variable. Initially, the independent variables in the model included age, sex, hypertension, diabetes, treatment delay, initial IHD, and recovery grade immediately after the treatment. Only the following three parameters significantly contributed and were included in the stepwise multiple regression model: age, initial IHD, and recovery grade IV. A *p*-value of less than 0.05 was considered statistically significant.

## 3. Results

### 3.1. Patient Information

In total, 473 patients visited the clinic for sudden SNHL, and 343 patients were excluded according to the inclusion and exclusion criteria. Six patients had underlying causes for sudden SNHL, such as vestibular schwannoma and temporal bone fracture, twenty-two patients were bilaterally affected, seventy-four patients were recurrently affected, and thirty-four patients had SNHL combined with vertigo, respectively. Sixty-seven patients were excluded because accurate audiometric information could not be achieved. They either lacked audiometric confirmation before treatment (38 patients) or had preceding asymmetric hearing loss (29 patients.) Twenty patients had more than 2 weeks of delay from symptom onset to initial treatment, and 120 patients were lost to follow-up before the end of treatment. Finally, 130 patients were included in the study (Figure 1). The patients’ demographic data is presented in Table 2.

### 3.2. Hearing Recovery Pattern According to the AAO-HNS Criteria

The hearing recovery pattern was analyzed immediately after treatment and 2 months after treatment completion. Immediately after treatment, 57 out of the 130 (43.9%) patients attained complete recovery (Grade I). Nineteen (14.6%) patients showed partial recovery but serviceable hearing (Grade II), and twenty-two patients (16.9%) showed Grade III recovery. Thirty-two (24.6%) patients attained less than 10-dB improvement (Grade IV, no recovery). The hearing recovery pattern 2 months after treatment completion showed a slight change compared to the hearing recovery distribution immediately after treatment. The number of Grade I patients increased from 57 to 60 (46.2%); two patients were previously Grade II, and one was Grade III. Five patients previously from Grade III and two patients from Grade IV improved to Grade II; thus, the number of Grade II patients also increased from 19 to 23 (17.7%). Seven patients from Grade IV improved to Grade III. However, the total number of Grade III patients was unchanged (22, 16.9%), but the number of Grade IV patients decreased from 32 to 25 (19.2%).

Out of the 130 patients with complete hearing recovery or a complete follow-up, the proportions of Grades I and II increased (2.4 and 3.1%), whereas Grade IV decreased (−5.4%). Moreover, 17 out of the 73 (23.3%) patients showed a grade promotion, 54 (74%) were constant, and 2 (2.7%) were aggravated. Changes in the hearing recovery patterns and mean hearing threshold are displayed in Figure 2 and Appendix A.

### 3.3. Delayed Hearing Recovery Pattern

Although 57 patients showed complete recovery after the initial treatment and were dismissed, 73 patients with incomplete/no recovery were reevaluated 2 months after treatment completion. An RM-ANOVA and multiple linear regression were conducted to estimate the degree of hearing recovery over time in 73 patients with incomplete hearing recovery. An RM-ANOVA with a Greenhouse–Geisser correction determined that the IHD differed statistically significantly between time points (F (1.270, 91.44) = 63.01, *p* < 0.001). The mean IHDs were 58.58 ± 26.25, 42.0 ± 23.84, and 36.96 ± 22.18 dB HL initially, immediately after treatment, and 2 months after treatment completion, respectively. Post hoc tests using Bonferroni correction revealed a significant hearing improvement from pretreatment to immediately after treatment (mean 16.58 dB HL, 95% confidence interval (CI) 11.14–22.01, *p* < 0.001) and immediately after treatment and 2 months after treatment completion (mean 5.041 dB HL, 95% CI 2.504–7.578, *p* < 0.001) (Figure 3).

Meanwhile, a stepwise multiple linear regression demonstrated the associated factors regarding the degree of delayed hearing recovery (F = 8.836, *p* < 0.001, adjusted R^2^ = 0.246). It was revealed that the initial IHD before the treatment, age, and the recovery grade immediately after treatment were significantly associated with a delayed recovery rate (Table 3, *p* < 0.001). When the other patient factors were controlled for, a more severe the initial hearing loss (*p* < 0.001), the older the age of the patient (*p* < 0.01), and worse patient recovery (*p* < 0.05) were found to be associated with a more delayed recovery. Other factors, such as sex, hypertension, diabetes, and the duration up to the treatment, did not significantly affect the delayed hearing improvement.

## 4. Discussion

There have been studies on the delayed recovery of ISSNHL in the past, but their patient groups and treatment methods were so heterogeneous that it was difficult to fully rely on their results. The present study applied strict diagnostic criteria to overcome those limitations. We provided a uniform treatment, not only the initial steroid administration but, also, salvage intratympanic steroid injection. We excluded patients with vertigo, because they might be diagnosed with other diseases in the future. We also tried to deduce the results more accurately by imposing actual IHD to measure the hearing changes and absolute auditory threshold level. With this highly balanced data of 130 patients, we investigated the long-term prognosis of ISSNHL. The main findings of this study can be summarized as follows. First, the hearing recovery continued for at least 2 months after the drug was no longer administered, and the degree of this delayed hearing recovery was statistically significant. Second, there were some factors that might affect a delayed hearing recovery, which were age, degree of initial hearing loss, and degree of recovery after initial treatment.

As discussed in many previous studies, there is an ongoing debate as to whether the recovery of hearing after the sudden hearing loss after treatment completion is due to the drugs or whether it is a natural recovery [14,15,16]. Similarly, it is difficult to determine whether the delayed recovery that occurs after discontinuation of the drug is also affected by the drug or not. A previous study reported that, among 215 patients with ISSNHL who did not receive the salvage therapy, only 6.5% developed delayed hearing recovery 1 month after drug discontinuation [17]. This is a significantly smaller number compared to those of the previous studies that received salvage therapy [18,19], including the present study. Considering this phenomenon, delayed recovery after drug discontinuation might not be a natural course but might be affected by the use of the drug. Well-designed further studies could provide an answer.

If drug use affects the delayed recovery of ISSNHL, the following hypothesis can be considered for the meaningful recovery of hearing after drug discontinuation. First, delayed recovery might be due to differences in etiologies. ISSNHL is idiopathic and can be due to various unknown causes. Certainly, different patterns of inner ear damage were confirmed in a histopathological study among patients diagnosed with ISSNHL [20]. If cochlear nerve damage prior to cochlea was dominant, the anti-inflammatory effects of corticosteroid would effectively prevent swelling of the cochlear nerve so that the hearing can be restored. However, if the cochlea was affected, the anti-inflammatory effect on the cochlear nerve would not affect the recovery of damaged hearing. This unknown difference in etiology may provide a possible explanation for the differences in the recovery time after corticosteroid treatment.

Second, the degree of damage to the auditory organ might affect the time it takes to recover. In the case of the facial nerve, the degree of damage is specified as neurotmesis, axonotmesis, and neuropraxia, and the prognosis is expected to some extent, according to the degree of damage [21]. Patients with a milder degree of Bell’s palsy are expected to have a higher probability of a complete recovery and shorter duration for recovery [22]. Currently, the only measure to evaluate the damage and recovery of the auditory organ in sudden hearing loss is the level of hearing. The main damage site and degree of damage remain unknown. Delayed recovery could be due to differences in the extent of damage of the auditory organs. In the present study, the patients with a poorer hearing level initially showed more delayed recovery, suggesting that more time was required for recovery when the auditory organs were severely damaged.

The present study aimed to investigate the long-term prognosis of ISSNHL with the highly balanced data of 130 patients. Nevertheless, this study had some limitations. First, given that the AAO-HNS guideline has recommended an observational period until 6 months after onset, the 2-month observational period after the treatment could be considered insufficient. However, our study was noteworthy in terms of its homogenous diagnosis, treatment, and surveillance period. Additionally, some studies have shown that there were no significant changes in hearing 2 months after treatment completion [23,24]. Second, the opportunity for selection bias remained due to the follow-up loss of alleviated patients; if the patients had a subjective recovery, they might not have visited the clinic, inducing a low recovery rate and significant results. Additionally, the exclusion of comorbid vertigo cases could have induced a higher recovery rate, since vertigo is associated with a worse outcome in ISSNHL [4]. We excluded those cases, considering that ISSNHL accompanied by vertigo might be the first symptom of Meniere’s disease. In the case of Meniere’s disease, the hearing recovery rate is reported to be higher than that of ISSNHL [25]. Third, the probable factors known to affect hearing recovery in ISSNHL have not been included for extensive analysis, and further comprehensive data collection is necessary.

## 5. Conclusions

The present study confirmed that a delayed recovery occurred within 2 months in more than 20% of patients with ISSNHL. A poorer hearing level at the time of onset might be a sign of a slower recovery rather than a poorer prognostic factor. The treatment outcome of ISSNHL should be evaluated at least 2 months after treatment completion, and counseling is required on the need for long-term follow-up in patients with ISSNHL.

## Figures and Tables

**Figure 1 jcm-11-02792-f001:**
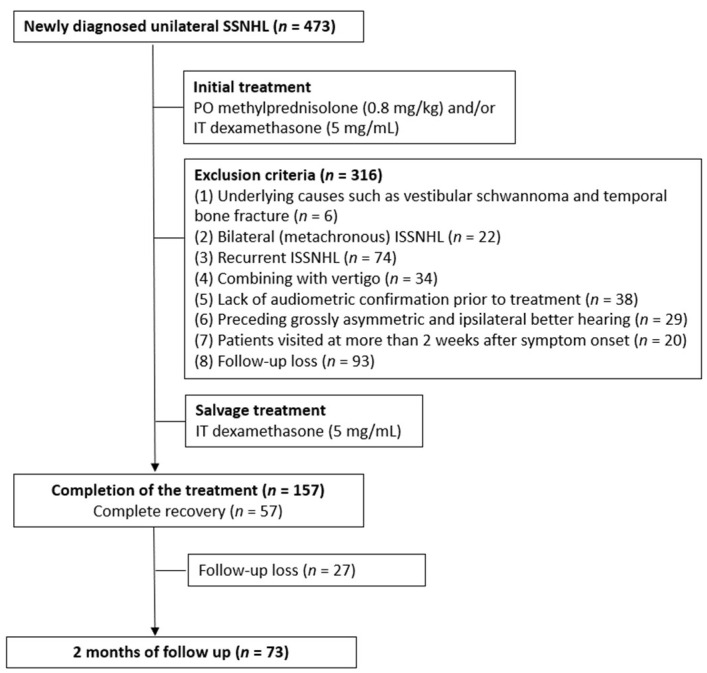
The diagram presents the number of subjects, exclusion criteria, and treatment protocols used in the study. ISSNHL, idiopathic sudden sensorineural hearing loss; IT, intratympanic; PO, peroral; SSNHL, sudden sensorineural hearing loss.

**Figure 2 jcm-11-02792-f002:**
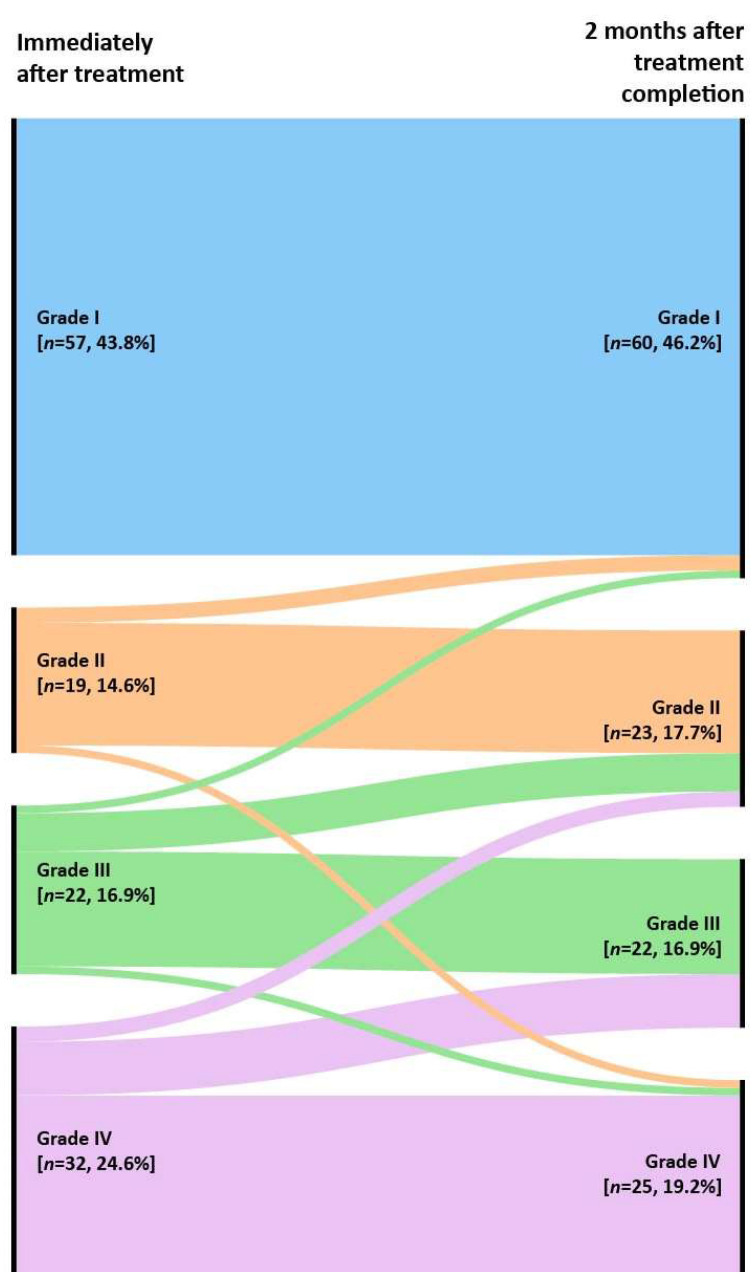
The alluvial diagram showed the distribution of the recovery grade from immediately after the treatment (left nodes) to 2 months after treatment completion (right nodes).

**Figure 3 jcm-11-02792-f003:**
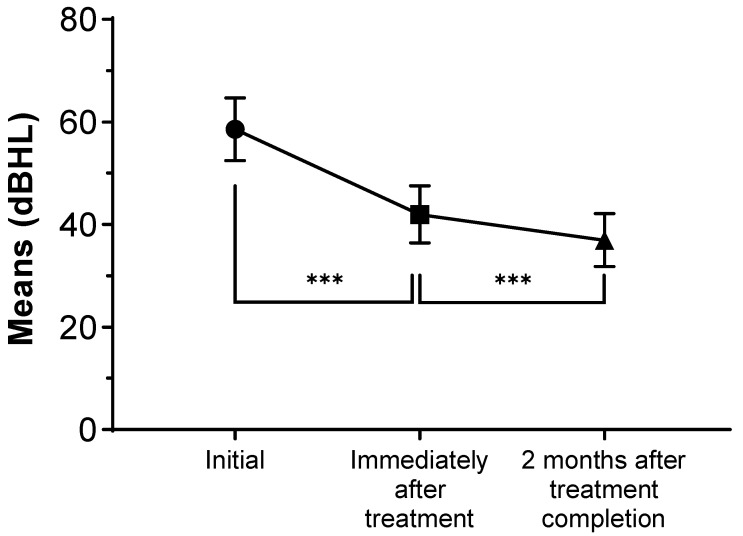
The repeated measures analysis of variance graph of the interaural hearing differences over time. The results are represented as the mean, and the vertical bars denote 95% confidence intervals. *** *p*-value less than 0.001.

**Table 1 jcm-11-02792-t001:** Grading system according to the pure-tone average, which was modified from the American Academy of Otolaryngology-Head and Neck surgery guideline.

Grade	Hearing Outcome
I. Complete recovery	PTA within 10 dB of the unaffected ear
II. Partial recovery, serviceable hearing	≥10 dB improvement in PTA, PTA ≤ 50 dB
III. Partial recovery, non-serviceable hearing	≥10 dB improvement in PTA, PTA > 50 dB
IV. No recovery	<10 dB improvement in PTA

PTA, pure-tone average.

**Table 2 jcm-11-02792-t002:** Patient demographics and audiologic data during the diagnosis of idiopathic sudden sensorineural hearing loss.

	*n*	Mean ± SD
Age	130	52.1 ± 16.3
Sex		
Male	78	40%
Female	52	60%
Site		
Right	56	43%
Left	74	57%
Hypertension	25	19%
Diabetes	23	18%
Days up to the treatment	130	3.57 ± 3.74
Initial hearing		
Ipsilateral PTA (dB HL)	130	67.4 ± 25.2
Contralateral PTA (dB HL)	130	16 ± 11.6
Interaural difference (dB HL)	130	51.3 ± 24.4
Ipsilateral WRS (%)	130	24.7 ± 33.2

PTA, pure-tone average; WRS, word recognition score.

**Table 3 jcm-11-02792-t003:** Stepwise multiple linear regression model predicting the association between delayed hearing gain and the independent variables (*n* = 73).

Effect		95% Confidence Interval			
	B	Lower	Upper	Beta	*p*-Value
Constant *	−17.038	−26.402	−7.674		0.001
Initial IHD (dB HL) *	0.182	0.106	0.259	0.541	<0.001
Age *	0.172	0.055	0.289	0.311	0.004
Recovery Grade 4 *	4.843	0.958	8.728	0.274	0.015
Sex				−0.089	0.399
HTN				0.138	0.197
DM				0.012	0.918
Days before the treatment				−0.070	0.521
Recovery Grade 3				−0.199	0.166

B = unstandardized coefficients; Beta = standardized coefficients; Sex: male = 1, female = 2; HTN, Hypertension: no = 0, yes= 1; DM, Diabetes: no = 0, yes= 1; IHD, interaural hearing difference. The recovery grade refers to the grade immediately after the treatment according to the pure-tone audiometry. Recovery grade 2 is the reference group. Reduced model: F = 8.836, *p* < 0.001, adjusted R^2^ = 0.246. * Variables included in the reduced model.

## Data Availability

The data are available from the corresponding author upon reasonable request.

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
