# Peer review of "Delayed Recovery in Idiopathic Sudden Sensorineural Hearing Loss"

_jcm, 2022, doi:10.3390/jcm11102792_

Round 1

Reviewer 1 Report

Na et al performed a retrospective study of patients with idiopathic sudden sensorineural hearing loss, a relatively common condition that remains incompletely understood. The study adds to the body of several similar studies already reported, which have also described ‘delayed recovery’. Nevertheless, the study includes well-described inclusion and exclusion criteria that may contribute prognostic data in select cases. However, the follow-up period was relatively short.

The etiopathology of idiopathic sudden sensorineural hearing loss remains incompletely understood, preventing development of better treatments, and may differ between patients. Nevertheless, it likely results from vestibulocochlear damage, which is sometimes irreversible and sometimes may recover. Steroids prevent ongoing damage and/or promote recovery, possibly by reducing inflammation and/or swelling.

Major points:

-The word ‘spontaneous’ should be removed from the title and discussion, as all patients received steroids. Steroids are known to promote recovery, but as nerve tissue regenerates at best slowly and there is no reason to believe that the kinetics of recovery should be linked to the pharmacokinetics of the drugs (discussion, lines 271-285).

-Vertigo may result as a consequence of severe vestibulocochlear lesions, and associate with worse outcome. This may explain why other studies (e.g. ref [17]) show less recovery but more grade 4 recovery. Please discuss.

-The attempt to perform a linear regression model is admirable but results need to be presented more clearly, it is impossible to comprehend the meaning of lines 201-209.

-There are several linguistic issues throughout the manuscript, only some of which are pointed out below. Please revise.

Minor:

-line 243: ‘delayed hearing loss’ -> ‘delayed hearing recovery’??

-line 256: ‘regarding sensory problem…’ ???

-line 263: explain ‘lower House-Brackmann grade’

-lines 291-292: change ‘proven that there were hardly significant changes’ -> ‘shown that there were no significant changes’

Author Response

Na et al performed a retrospective study of patients with idiopathic sudden sensorineural hearing loss, a relatively common condition that remains incompletely understood. The study adds to the body of several similar studies already reported, which have also described ‘delayed recovery’. Nevertheless, the study includes well-described inclusion and exclusion criteria that may contribute prognostic data in select cases. However, the follow-up period was relatively short.

The etiopathology of idiopathic sudden sensorineural hearing loss remains incompletely understood, preventing development of better treatments, and may differ between patients. Nevertheless, it likely results from vestibulocochlear damage, which is sometimes irreversible and sometimes may recover. Steroids prevent ongoing damage and/or promote recovery, possibly by reducing inflammation and/or swelling.

Response: We would like to thank the reviewer for the thorough and helpful review of the manuscript. We agree with reviewer’s comments and opinions, and have made changes to the manuscript accordingly. We believe that these changes make the paper more persuasive and informative. Below is the point-by-point response to the reviewer’s comments.

Major points:

-The word ‘spontaneous’ should be removed from the title and discussion, as all patients received steroids. Steroids are known to promote recovery, but as nerve tissue regenerates at best slowly and there is no reason to believe that the kinetics of recovery should be linked to the pharmacokinetics of the drugs (discussion, lines 271-285).

Response: Thank you for your valuable comment. The authors agree that the word ‘spontaneous’ could lead to confusion to the readers that there was no treatment at all, but the intentional meaning was confined only to the period ‘after the steroid cessation’. We have removed the word from the title and throughout the manuscript. We also have deleted the paragraph in the Discussion section regarding the correlation between pharmacokinetics and recovery rate.

-Vertigo may result as a consequence of severe vestibulocochlear lesions, and associate with worse outcome. This may explain why other studies (e.g. ref [17]) show less recovery but more grade 4 recovery. Please discuss.

Response: Thank you for your precious comment. Vertigo is one of the well-known poor prognostic factors in recovery of sudden hearing loss. The authors excluded the subjects accompanied by vertigo in the present study, and as the reviewer pointed out, that could have brought the higher recovery rate as compared to other studies. On the other hand, sudden hearing loss accompanied by vertigo might be an initial event of future Meniere’s disease. Most studies agreed that the recovery rate of hearing loss is higher in patients with Meniere’s disease than in those with idiopathic sudden hearing loss. As the authors hoped to focus on the capability of delayed recovery in idiopathic sudden hearing loss, which might not exceed the uncertain effect of another unrevealed disease, we initially excluded the comorbid vertigo cases in the study. Accordingly, we have discussed it in the revised manuscript.

(lines 299-303) Also, exclusion of comorbid vertigo cases could have induced higher recovery rate, since vertigo is associated with worse outcome in ISSNHL. We excluded those cases, considering that ISSNHL accompanied by vertigo might be the first symptom of Meniere’s disease. In case of Meniere’s disease, hearing recovery rate is reported to be higher than that of ISSNHL.

-The attempt to perform a linear regression model is admirable but results need to be presented more clearly, it is impossible to comprehend the meaning of lines 201-209.

Response: Thank you for pointing this out. We have revised the description for better understanding.

(lines 201-211) Meanwhile, a stepwise multiple linear regression demonstrated the associated factors regarding the degree of delayed hearing recovery (F = 8.836, p < 0.001, adjusted R2 = 0.246). It was revealed that initial IHD before the treatment, age, and the recovery grade immediately after treatment were significantly associated with the delayed recovery rate (Table 3, p < 0.001). When the other patient factors were controlled for, more severe the initial hearing loss (p < 0.001), older age of the patient (p < 0.01), and worse patient recovery (p < 0.05) were found to be associated with a more delayed recovery. Other factors, such as sex, hypertension, diabetes, and the duration up to the treatment did not significantly affect the delayed hearing improvement.

-There are several linguistic issues throughout the manuscript, only some of which are pointed out below. Please revise.

Response: Thanks for pointing these out. We have corrected the addressed issues appropriately. Also, the original and revised versions of the manuscript have been proofread by professional native English language editors. Please find attached our certificate of English editing at the end of the letter.

Minor:

-line 243: ‘delayed hearing loss’ -> ‘delayed hearing recovery’??

Response: Thank you for pointing this out. We have revised the word.

-line 256: ‘regarding sensory problem…’ ???

Response: Thank you for pointing this out. We have modified the paragraph for better understanding as follows.

(lines 256-262) If cochlear nerve damage prior to cochlea was dominant, the anti-inflammatory effects of corticosteroid would effectively prevent swelling of the cochlear nerve so that the hearing can be restored. However, if the cochlea was affected, anti-inflammatory effect to the cochlear nerve would not affect the recovery of damaged hearing. This unknown difference in etiology may provide a possible explanation for the differences in the recovery time after corticosteroid treatment.

-line 263: explain ‘lower House-Brackmann grade’

Response: Thank you for your comment. We have revised the description as ‘milder degree of Bell’s palsy’.

-lines 291-292: change ‘proven that there were hardly significant changes’ -> ‘shown that there were no significant changes’

Response: Thank you for pointing this out. We have revised the sentence.

Reviewer 2 Report

I would like to thank the Editor for the invitation to review this paper. The authors present an interesting work on the long-term prognosis of idiopathic sudden sensorineural hearing loss (ISSNHL). They use rigorous inclusion and exclusion criteria to obtain a homogeneous sample. I especially consider the definition of a clear treatment protocol, with oral or intratympanic corticosteroids for all patients, and intratympanic injection also being a salvage option.

The main innovation of the present study with respect to previous studies is the application of strict diagnostic criteria and uniform treatment.

Although the follow-up period is clinically interesting, I miss another control at 6 months to assess the evolution of patients without recovery or with partial recovery at 2 months. I think it would add relevant information.

I consider this paper is interested in the subject for general practitioners and otolaryngologists, as well as for providing relevant data on ISSNHL.

Minor changes suggested:

Line 243: “only 6.5% developed delayed hearing loss -recovery- 1 month after drug…”

Author Response

I would like to thank the Editor for the invitation to review this paper. The authors present an interesting work on the long-term prognosis of idiopathic sudden sensorineural hearing loss (ISSNHL). They use rigorous inclusion and exclusion criteria to obtain a homogeneous sample. I especially consider the definition of a clear treatment protocol, with oral or intratympanic corticosteroids for all patients, and intratympanic injection also being a salvage option.

The main innovation of the present study with respect to previous studies is the application of strict diagnostic criteria and uniform treatment.

Although the follow-up period is clinically interesting, I miss another control at 6 months to assess the evolution of patients without recovery or with partial recovery at 2 months. I think it would add relevant information.

I consider this paper is interested in the subject for general practitioners and otolaryngologists, as well as for providing relevant data on ISSNHL.

Response: We would like to thank the reviewer for thorough and helpful review of the manuscript. As you mentioned, it is unfortunate that we presented the results after only two months. We considered this to be a limitation of our study and discussed it. We will try to find any changes at six months compared to two months in our future study.

Minor changes suggested:

Line 243: “only 6.5% developed delayed hearing loss -recovery- 1 month after drug…”

Response: Thank you for pointing this out. We have corrected the word.

Reviewer 3 Report

The study investigates the factors that may influence the poor outcomes of sudden deafness. The study is well designed, and the article is well written. However, numerous prior studies have published similar results, there is a lack of novelty in this paper. An ideal design should put more parameters in the model to study a much bigger population.   

Author Response

The study investigates the factors that may influence the poor outcomes of sudden deafness. The study is well designed, and the article is well written. However, numerous prior studies have published similar results, there is a lack of novelty in this paper. An ideal design should put more parameters in the model to study a much bigger population.

Response: Thank you for this thoughtful comment. As you mentioned, there were numerous prior studies. However, many past studies were lacking a clear definition of uniform protocol or patient population. Currently, although guideline-supported medicine is only corticosteroids with proper duration, in actual clinical practice, various treatments with insufficient evidence are being implemented, probably due to the inherent limitations of “idiopathic” sudden hearing loss. Our study tried to exclude these confounding factors as much as possible and focused on the period after treatment, which is easily overlooked. Even though the patient population decreased by 85% of the whole patients due to the strict inclusion and exclusion criteria, we believe that this clearly organized study will provide useful information for patient counseling at the real clinician's clinic.
